# Dietary Exposures and Intake Doses to Bisphenol A and Triclosan in 188 Duplicate-Single Solid Food Items Consumed by US Adults

**DOI:** 10.3390/ijerph18084387

**Published:** 2021-04-20

**Authors:** Marsha K. Morgan, Matthew S. Clifton

**Affiliations:** United States Environmental Protection Agency, 109 T.W. Alexander Drive, Research Triangle Park, NC 27711, USA; clifton.matthew@epa.gov

**Keywords:** bisphenol A, triclosan, consumer products, diet, exposure, adults

## Abstract

Few data exist on bisphenol A (BPA) or triclosan (TCS) residue levels in foods consumed by adults in everyday settings. In a further analysis of study data, the objectives were to determine BPA and TCS residue concentrations in duplicate-single solid food items consumed by adults and to estimate dietary exposure and intake doses per food item. A convenience sample of 50 adults was recruited in North Carolina (2009–2011). Participants completed 24 h food diaries and collected 24 h duplicate-diet solid food samples consumed on days 1 and 2 during sampling weeks 1, 2, and 6. A total of 188 of the collected 776 duplicate-diet solid food samples contained a single, solid food item. BPA and TCS residue levels were quantified in the 188 food items using GC–MS. BPA and TCS were detected in 37% and 58% of these food items, respectively. BPA concentrations were highest in a cheese and tomato sandwich (104 ng/g), whereas the highest TCS concentrations were in a burrito (22.1 ng/g). These chemicals co-occurred in 20% of the samples (maximum = 54.7 ng/g). Maximum dietary intake doses were 429 ng/kg/day for BPA in a vegetable soup with tortilla sample and 72.0 ng/kg/day for TCS in a burrito sample.

## 1. Introduction

Bisphenol A (BPA) is a synthetic phenol with over 2 billion pounds used each year in the United States (US) [1,2]. Research has indicated that BPA is an endocrine disruptor and may be causing adverse health effects (e.g., reproductive, neurological, and obesogenic) in exposed humans [1,2]. Since the 1960s, BPA is mainly used as a monomer in the production of light-weight polycarbonate plastics (75%) and epoxy resins (20%) [2,3]. Polycarbonate plastics are commonly found in a variety of consumer goods that are made to hold or store foods or beverages such as reusable storage containers, tableware, water bottles, plastic wrap, and microwave ovenware [3,4]. For epoxy resins, they are often applied as a coating to the inside of metal food cans and beverage cans, including caps, to prevent rust [4,5]. Several studies have reported that BPA can migrate into foods and beverages from some consumer goods, especially from metal cans [1,6,7]. Currently, the US Food and Drug Administration (FDA) does not regulate the amount of BPA that can occur in commercially available foods and beverages [8,9]. 

Previous studies have quantified BPA residue concentrations mainly in food items that were purchased from grocery stores and supermarkets and then analyzed in laboratories [7,8,10,11,12,13,14]. Little research has been conducted on the levels of BPA residues in the actual foods prepared and/or consumed by adults in their everyday environments [9,15,16]. These types of studies are necessary as research has indicated that people’s personal behaviors (i.e., microwaving food in containers made of BPA, consuming the liquid portion of canned vegetables, and not washing hands prior to preparing foods) can increase BPA residue levels in some foods prior to consumption [7,17,18,19]. Only one published study (“The Pilot Study to Estimate Human Exposures to Pyrethroids using an Exposure Reconstruction Approach [Ex-R study]”) has measured BPA residue concentrations in the actual foods consumed by adults in real-world settings [9]. In that study conducted in the period 2009–2011, BPA residues were detected in more than 37% of the duplicate-diet solid food samples (*n* = 776) consumed by 50 North Carolina (NC) adults in their everyday environments (i.e., home, work, and school). The maximum BPA residue level was 138 ng/g in the Ex-R food samples.

Triclosan (TCS) is a chlorinated phenol with more than 1 million pounds produced annually in the US [20,21,22]. Studies have shown that TCS is an endocrine disruptor and may be impacting the health (i.e., thyroid and reproductive) of exposed humans [20,21]. This phenol is not currently used in any food packaging materials (e.g., boxes, cans, bags, or cartons) in the US [22] and in countries from the European Union [21] However, in the last 30 years, TCS has been increasingly used as an antimicrobial agent in many other types of consumer goods including hand sanitizers, hand soaps, bar soaps, body washes, dishwashing liquids, toothpaste, deodorants, lotions, cosmetics, and kitchen cutting boards, utensils, and sponges [21,22,23,24]. The amount of TCS added to these consumer goods generally ranges between 0.1–0.3% by weight [25]. In 2016, the US FDA banned the use of this chemical in all hand soaps, bar soaps, and body washes because these personal care products were not any more effective at killing germs than using plain soap and water [26]. Nevertheless, concerns have been recently raised about the possible dietary exposures of adults to this lipophilic phenol via the indirect use of other commercially available consumer goods (e.g., cutting boards, hand sanitizers, skin lotions, dishwashing liquids, and napkins) [21,22,27,28]. 

Few published studies have measured the levels of TCS residues in any food samples, worldwide [28,29,30,31]. In a small study, Yao et al. [29] reported finding TCS residues in 60% of the fresh chicken eggs (maximum = 6.7 ng/g) bought from local supermarkets in Beijing, China [29]. More recently, Morgan and Clifton [31] detected TCS residues in 59% of the 776 duplicate-diet solid food samples (maximum = 394 ng/g) consumed by 50 Ex-R study adults in NC [31]. The authors suggested that the adult’s use of certain types of consumer goods (e.g., personal care products on skin or kitchenware) containing TCS while making and/or eating food items may have substantially elevated TCS levels in some foods before consumption [22,31]. 

In our previous work from the Ex-R study [9,31], we quantified the concentrations of BPA and TCS in 776 duplicate-diet solid food samples of 50 adults and estimated their maximum dietary exposures and dietary intake doses to these two chemicals. However, at that time, it was unclear what were the specific consumed solid food items (e.g., cheeseburger, salad, and pizza) that were likely substantially contributing to the dietary exposures of the Ex-R adults to BPA or TCS. In a further analysis of the Ex-R study data, the participants’ food diary records were used to determine that 188 of the collected 776 duplicate-diet solid food samples contained only a single food item. For this current work, the main objectives were to (1) determine BPA and TCS residue concentrations in the 188 duplicate-single solid food items consumed by adults and (2) estimate dietary exposure and dietary intake doses per food item. This research is important as we are unaware of any published study that has reported BPA or TCS residue concentrations in duplicate-single solid food items consumed by adults in their everyday settings.

## 2. Materials and Methods 

### 2.1. Study Background

The Ex-R study was designed to investigate adult exposures to several, current use pyrethroid insecticides in residential settings [32]. This observational exposure measurements study was conducted at the US Environmental Protection Agency’s (EPA) Human Studies Facility (HSF) in the state of NC and at the participant’s residences (within a 40-mile radius of the HSF) in the period 2009–2011. As part of this study, 50 adult participants, aged 19–50 years old, completed 24 h food diaries and collected 24 h duplicate-diet solid food samples on days 1 and 2 of each sampling week (1, 2, and 6) over a six-week monitoring period. For each sampling day, the individual participants collected up to three separate duplicate-diet solid food samples. The Ex-R participants collected a total of 776 duplicate-diet solid food samples during the monitoring period.

At a US EPA laboratory in NC in the period 2015–2016, the duplicate-diet solid food samples (*n* = 776) were analyzed for additional target chemicals (BPA, several BPA analogues [B, F, P, S, and Z], and triclosan) that are commonly found in consumer goods [31]. Of these chemicals, only BPA and TCS were detected in ≥38% of the duplicate-diet solid food samples. For this current work, we have quantified BPA and TCS residue levels occurring in the 188 duplicate-diet solid food samples that contained only a single food item. 

### 2.2. Human Subjects Protection

The US EPA’s Human Subjects Research Review Official and the University of NC’s Institutional Review Board approved the protocol and procedures for the Ex-R study (study number 09-0741) in 2008 [32]. Adult volunteers signed a paper copy of the informed consent form before participating in the Ex-R study. The adults were also assigned unique participant identification numbers to protect their privacy. 

### 2.3. Collection of the Food Diaries and Solid Food Items

The collection of the 24 h food diaries by the adult participants was described previously in Morgan et al. [32]. Briefly, the participants completed hardcopies of the 24 h food diaries on days 1 and 2 of each sampling week (1, 2, or 6). The 24 h food diaries were filled out over three, consecutive sampling periods each sampling day. The three sampling times consisted of period 1 [4:00–11:00 am], period 2 [11:00 am–5:00 pm], and period 3 [5:00 pm–4:00 am]. For each sampling period, the food diary was used to record each single, solid food item (i.e., burrito, cereal and milk, oatmeal with banana, cheeseburger, and salad with chicken) that was eaten by a participant. Additionally, in this food diary, the participant checked a box to record that each solid food item they consumed was also part of the corresponding duplicate-diet solid food sample for this sampling period. 

The collection of the 24 h duplicate-diet solid food samples by the participants on days 1 and 2 of each sampling week was mentioned previously in Morgan et al. [32]. In short, duplicate-diet solid food samples were defined as identical quantities of all the solid foods items (no beverages) that were purchased and eaten by a participant during each sampling period [32]. The 24 h duplicate-diet solid food samples were collected by each participant over three, consecutive sampling periods each sampling day (as stated above). The participants put duplicate quantities of all solid food items that they consumed for each sampling period into the same, resealable sampling bag made of 100% polyethylene (31 × 31 cm, Uline Shipping Supply Specialist^®^, Pleasant Prairie, WI, USA). Each food sampling bag was placed into a portable, light-weight thermoelectric cooler with wheels (34 cm L x 30 cm W x 36 cm H, Princess International^®^ or Vinotemp^®^, Los Angeles, CA, USA) [32]. 

On day 3 of each sampling week, the participants’ transported (i.e., by car) the portable coolers with the food diaries and food sampling bags to the HSF. At the HSF, a researcher verified with the individual participants that the food diaries were properly filled out and all duplicate quantities of consumed solid food items had been placed into the correct sampling bags [32]. Additionally, the condition of each sampling bag (e.g., no leaks) was noted, and the mass (g) of each sampling bag was recorded with a calibrated, weight scale. Using a van, a researcher transported the portable coolers with the food diaries/samples (with ice) to a US EPA laboratory in Durham, NC [32]. At the laboratory, for each food sampling bag, all food item(s) were homogenized together using a vertical cutter mixer or blender and then aliquoted (12.0 g each) into individual amber glass jars (30 mL) [31]. The glass jars were kept in freezers (≤−20 °C) until analysis in the period 2015–2016 [31]. 

### 2.4. Chemical Analysis of the Solid Food Items

A modified QuEChERS method (quick, easy, cheap, effective, rugged, and safe) was used to extract BPA and TCS residues from the solid food samples (*n* = 188) [31,33]. For each food sample, a glass jar containing 12.0 g of homogenized food was removed from a freezer and thawed overnight in a refrigerator at the laboratory. From each glass jar, 2 g of a food sample was placed into a 50 mL polypropylene centrifuge tube. Each sample was fortified with an internal standard solution prior to the addition of a ceramic homogenizer bar and 12 mL of acetonitrile. The tubes were capped and vortexed for 1 min, allowed to sit for 1 min, and then vortexed again (1 min). Then, an extraction salt (0.8 g MgSO_4_ with 0.2 g sodium acetate) was added to each tube and vortexed for 1 min. The tubes were then centrifuged for 5 min at 4000 RPM; the acetonitrile layer was transferred to 15 mL polypropylene centrifuge tubes containing 0.4 g of graphitized carbon black. These tubes were capped, vortexed for 1 min, and centrifuged for 5 min at 4000 RPM. The acetonitrile layer was transferred to 50 mL glass centrifuge tubes. Then, the extracts were evaporated just to dryness using a Multivapor P-6 parallel evaporator (Buchi Labortechnik AG, Flawil, Switzerland). Samples were re-constituted with 1 mL of acetonitrile and vortexed for 1 min. Lastly, the re-constituted extracts were transferred to autosampler vials and silylated with 100 µL of Sylon BFT (Supelco, Bellefonte, PA, USA) at 80 °C for 15 min. 

Prior to sample analysis, the extracts were vortexed (~15 s). The extracts were quantified for BPA and TCS residue concentrations using gas chromatography/mass spectrometry (Agilent Technologies, Palo Alto, CA, USA) [31]. The method quantifiable limit (MQL) was 1.1 ng/g for BPA and 0.3 ng/g for TCS. The method detection limit (MDL) was also 1.1 ng/g for BPA, but slightly lower at 0.21 ng/g for TCS. The quality assurance and quality control (QC) procedures used for the food samples was described earlier in Morgan and Clifton [31]. The QC samples consisted of matrix blanks, matrix spikes, recovery spikes, and reagent blanks, and the QC results for these samples met all data quality acceptance criteria [31].

### 2.5. Statistical Analysis

Data values below the MQL for BPA or TCS were replaced by the value of MQL/√2. Descriptive statistics including detection frequency, median, percentiles [75th and 95th], and range, were computed for BPA and TCS residue levels in the 188 food items that were consumed by 46 different Ex-R participants (JMP software, 2015 version 12.0.1, SAS Institute Inc., Cary, NC, USA). Using the method described in Morgan [19], the dietary exposure (Equation (1)) and dietary intake dose (Equation (2)) to BPA or TCS in each participant’s consumed food item were calculated as follows:E = C × M(1)
where E equals a person’s dietary exposure (ng/day) to a chemical, C equals the concentration (ng/g) of a chemical in a food item, and M equals the mass of food (g) in the sampling bag.
D = EA/B(2)
where D equals a person’s dietary intake dose (ng/kg/day) to a chemical, E equals the dietary exposure estimate (ng/day), A equals a default absorption rate of 100% in the gastrointestinal tract [1,20], and B equals body weight (kg). Food mass (g) records were missing for 28 food items because the weights of the sampling bags were not recorded for the first eight adults in this study. The missing food masses for these samples were estimated by using the corresponding amounts recorded (in cups) in the food diaries for these participants. 

## 3. Results

### 3.1. Residue Levels of BPA and TCS in the Food Items

BPA was detected in 37% of the 188 duplicate-single, solid food samples consumed by the Ex-R participants. Residue levels of BPA in all the food samples were 1.6 ng/g at the 75th percentile and 16.7 ng/g at the 95th percentile. Table 1 presents the concentrations of BPA residues that were detectable (≥MQL) in each food item. For these food samples, the median BPA residue level was 2.4 ng/g and at the 95th percentile it was 53.3 ng/g. The results showed that 17% of these samples had BPA residue levels above 11 ng/g. The highest BPA residue level was found in a cheese and tomato sandwich (104 ng/g) followed by a vegetable soup with tortilla (63.1 ng/g). BPA residues were found the most often in various types of sandwiches (range = 1.4–104 ng/g) and yogurts with cereals (1.1–3.2 ng/g).

TCS residues were detected in 58% of the 188 duplicate-single, solid food items consumed by the participants. For all the food samples, the median TCS residue level was 0.6 ng/g (95th percentile = 8.3 ng/g). Table 2 provides the concentrations of TCS residues that were detectable (≥MQL) in each food item. For these samples, median TCS residue levels were 1.7 ng/g and 13.3 ng/g at the 95th percentile. The greatest levels of TCS were found in a burrito sample (22.1 ng/g) and a bacon and egg sandwich (17.4 ng/g). Residues for this chemical occurred the most often in various kinds of sandwiches (range = 0.6–17.4 ng/g).

### 3.2. Co-Occurrence of BPA and TCS in the Food Items

BPA and TCS residues co-occurred in 20% of all the food items. Figure 1 presents the 37 different food samples that had detectable levels (≥MQL) of both BPA and TCS. The highest combined residue levels of these two chemicals were in a ham and cheese sandwich (54.7 ng/g) followed by a bowl of vegetable soup (34.0 ng/g). These two chemicals co-occurred the most often in various types of sandwiches (range = 3.0–54.7 ng/g), particularly those containing eggs, cheese and/or ham. Other foods that these two chemicals co-occurred most frequently in were cereals with milks (range = 1.4–7.9 ng/g) and pizzas (range = 3.9–6.9 ng/g). The results showed that 59% of these food items contained higher residue concentrations of BPA compared to TCS (Figure 1).

### 3.3. Estimated Dietary Exposure and Dietary Intake Doses to BPA and TCS per Food Item

Table 3 presents the adults’ estimated dietary exposures and dietary intake doses to BPA for each food item (≥MQL). For these samples, the participant’s median dietary exposure to BPA was 499 ng/day (95th percentile = 12,920 ng/day), and the median dietary intake dose to BPA was 6.0 ng/kg/day (95th percentile = 194 ng/kg/day). The maximum dietary intake dose of 429 ng/kg/day occurred in a female participant’s vegetable soup with tortilla sample that was eaten inside her home. Additionally, the second highest dietary intake dose was in another vegetable soup sample (261 ng/kg/day) consumed by a male participant indoors at home.

For TCS, the adult’s estimated dietary exposures and dietary intake doses for each food item (≥MQL) are provided in Table 4. The median dietary exposure to TCS was 345 ng/day (95th percentile = 2557 ng/day), and the median dietary intake dose to TCS was 4.2 ng/kg/day (95th percentile = 29.5 ng/kg/day). The maximum dietary intake dose was found in a male participant’s burrito sample (72.0 ng/kg/day) eaten while in transit (e.g., car).

## 4. Discussion

In the last decade, several published studies have reported finding measurable levels of BPA residues in a variety of food items purchased from grocery stores and supermarkets around the world [4,8,10,11,12,13,14,34]. These prior studies found the highest BPA residue concentrations occurring mainly in foods that originated from metal cans (i.e., vegetables, meats, and soups). In the US, Lorber et al. [13] detected BPA residues in 27% of the 116 food items (maximum = 149 ng/g) collected at grocery stores in Texas in 2010. Of these samples, BPA residues were found in 70% of the canned foods and 6% of the non-canned foods [13]. In comparison to these earlier studies, our current work quantified the concentrations of BPA residues in 188 duplicate-single, solid food items that were collected by 50 Ex-R adults in their everyday environments in the period 2009–2011. BPA residues were detected in 37% of these food samples with the highest residue level of 104 ng/g occurring in a participant’s cheese and (fresh) tomato sandwich. This chemical was found the most often in prepared sandwiches (maximum = 104 ng/g), particularly ones containing eggs, cheese, and/or ham. In addition, the top four out of six food samples that contained the greatest BPA residue levels were various kinds of sandwiches and bagels (Table 3). In the US, components of these foods (i.e., bread, cheese, raisins, ham slices, pickles, fresh tomatoes and/or lettuce) used to make these sandwiches/bagels are typically stored in packaging made of plastic, cardboard, and/or glass (not metal cans). An interesting observation was that a female participant (in her early 50s) ate three of these top food items including a cheese and tomato sandwich (104 ng/g/) and two different raisin and tomato bagels (42.9 ng/g and 45.7 ng/g) while inside her home. Other food samples she consumed indoors at home including toast, a raisin bagel, and a cheese and tomato bagel, contained much lower BPA residues (<3.7 ng/g). As sandwiches and bagels are commonly prepared and/or eaten by hand, it is suspected that her unknown personal behavior (e.g., dermal use of personal care products) likely contributed to her dietary exposure to BPA in some of these consumed food samples. This is possible as Liao and Kannan [23] reported finding this lipophilic chemical in several different types of personal care products including body washes, make up, hair care products, skin lotions, and toilet soaps in the US and China in 2012–2013. In a more recent study, Lu et al. [35] also reported finding BPA in body lotions (25%), lipsticks (100%), hand lotions (53%), shampoos (20%), facial masks (73%), and sunscreens (42%) purchased from grocery and retail stores in China in 2015. Furthermore, in a novel study by Hormann et al. [36] they reported adults that used hand sanitizer before handling cash register receipts (made of BPA) had a significant amount of this chemical transferred to their hands and subsequently to consumed food (French fries). As scant data are available, more research is needed on the dermal (hand) transfer of BPA from consumer goods, including personal care products, to foods prior to consumption.

Few data are available on the occurrence of TCS residue concentrations in foods commonly consumed by adults in everyday settings, globally [31]. In this current work, TCS residues were detected in 58% of the 188 duplicate-single, solid food items (maximum = 22.1 ng/g). Residue levels were found the most often in prepared sandwiches (maximum = 17.4 ng/g), especially ones containing eggs, cheese, and/or ham. Additionally, the top 9 out of 10 food items with the greatest TCS residues (≥7.7 ng/g) were most likely eaten by the Ex-R adults using their hands (Table 4). In addition, five of these top items, all sandwiches, were consumed by the same female participant (in her mid-20s) inside her home. These items were a bacon and egg sandwich (17.4 ng/g), a fried ham and egg sandwich (14.5 ng/g), and three different ham and cheese sandwiches (range = 7.7–15.4 ng/g). As there are no known uses of TCS in food packaging in the US [22], this information suggests that people’s use of other consumer goods (i.e., personal care products or kitchenware) containing TCS may be indirectly contributing to their dietary exposures to this chemical. During the time of the Ex-R study, this lipophilic chemical was frequently used as an active ingredient in >92% of the antibacterial liquid, gel, and foam soaps in the US [37,38]. Additionally, Liao and Kannan [23] reported finding TCS residues in several different types of personal care products purchased from retail stores in the US and China in 2012–2013. In that study, the greatest maximum TCS residues were in commonly used body washes (53,900 ng/g) and skin lotions (2890 ng/g). In addition, Canosa et al. [28] showed that TCS residues were transferrable from a commercial kitchen cutting board, impregnated with TCS, to foods (cheese [50 ng/g] and boiled ham [40 ng/g]) within five minutes of contact. This above information suggests that people’ use of some consumer goods containing this chemical is likely contaminating some foods prior to consumption, and more research is warranted. 

No published studies have reported the co-occurrence of BPA and TCS residues in single, solid food items consumed by adults. In this current work, these two chemicals co-occurred in 20% of the 188 solid food items consumed by Ex-R adults (Figure 1). BPA and TCS were found together the most often in prepared sandwiches (maximum = 54.7 ng/g). Foods that had combined residues of BPA and TCS above 20 ng/g included a burrito, chicken wrap with vegetables, vegetable soup, a ham and egg sandwich, and two different ham and cheese sandwiches. As few data exist, additional research is needed on the co-occurrence of consumer chemicals like BPA and TCS in foods commonly eaten by people in everyday settings.

In previous work, the Ex-R adult’s maximum dietary intake doses to BPA and TCS over a 24 h period were estimated to be 0.77 µg/kg/day and 1.6 µg/kg/day, respectively [9,31]. These values were well below the US EPA’s established oral reference doses (RfDs) of 50 µg/kg/day for BPA and 300 µg/kg/day for TCS [22,39]. However, at that time, it remained unknown what were the solid food items that likely contributed the most to the participants’ estimated dietary intake doses to these two chemicals. In this current study, the Ex-R adults’ highest dietary intake doses of BPA were in a vegetable soup with tortilla (0.43 µg/kg/day) sample followed by another vegetable soup sample (0.26 µg/kg/day). These data agree with Lorber et al. [13] who collected 116 foods from US grocery stores in Texas in 2010 and reported that people’s dietary intake doses of BPA were mainly driven by consumption of foods containing vegetables (94%) originating from metal cans. However, our adult dietary intake dose data also indicated that prepared sandwiches and bagels were substantial (non-canned) sources of BPA in the diet (Table 3). For instance, the estimated dietary intake dose for a participant’s cheese and fresh tomato sandwich was 0.24 µg/kg/day. For TCS, the Ex-R participants’ greatest intake doses were for a burrito (0.072 µg/kg/day), stir fry with sesame noodle (0.039 µg/kg/day), vegetable soup (0.039 µg/g), and a buffalo chicken sandwich (0.038 µg/kg/day). It remains unclear why the highest dietary intake doses for these participants were found in these specific consumed food samples. Nevertheless, these data are significant as they show that dietary ingestion is likely a major route of exposure to TCS for adults in their daily environments. 

## 5. Conclusions

Based on the food sample data, the Ex-R participants were exposed to measurable levels of BPA and TCS residues in a variety of food items that they ate in their everyday lives. Our results indicate that foods typically prepared and/or consumed by hand (i.e., sandwiches) had some of the highest levels of BPA and TCS residues found in this study. This information suggests that the dermal (hand) transfer of BPA and TCS from some consumer goods (e.g., personal care products) to foods may be a significant pathway of exposure in the diet.

## Figures and Tables

**Figure 1 ijerph-18-04387-f001:**
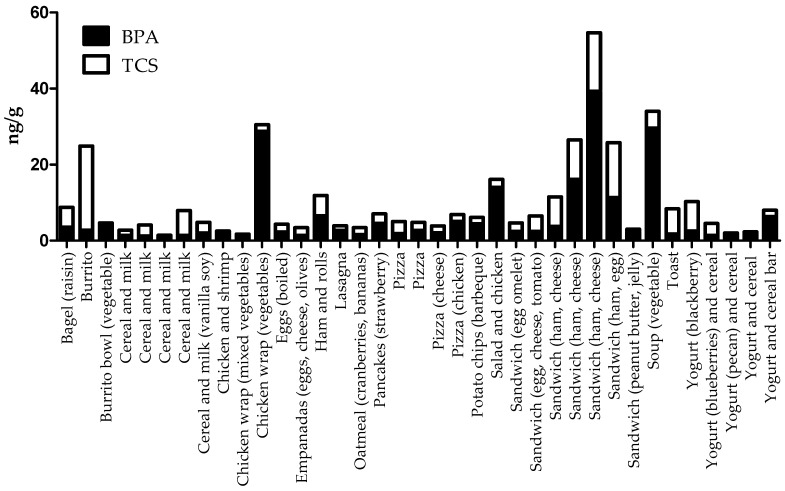
BPA and TCS residues co-occurred in 37 different solid food items.

**Table 1 ijerph-18-04387-t001:** Food items (*n* = 69) that contained detectable levels of BPA residues ^a^.

Food Item	BPA (ng/g)	Food Item	BPA (ng/g)
Applesauce (strawberry)	2.4	Pizza	2.7
Bagel (blueberry, cream cheese)	1.6	Pizza (cheese)	2.1
Bagel (cheese, tomato)	2.7	Pizza (chicken)	5.1
Bagel (raisin)	3.6	Potato chips (barbeque)	4.4
Bagel (raisin, tomato)	45.7	Salad (Caesar)	1.4
Bagel (raisin, tomato)	42.9	Salad (chicken barbeque)	5.6
Biscuit (sausage)	1.8	Salad and chicken	14.0
Burrito	2.8	Salad (goat cheese, toast)	17.2
Burrito (breakfast style)	1.5	Sandwich (cheese, tomato)	104
Burrito bowl (vegetable)	4.1	Sandwich (egg)	1.5
Cake (chocolate)	1.6	Sandwich (egg omelet)	2.4
Cereal with milk	1.1	Sandwich (egg, cheese, tomato)	2.4
Cereal and milk	1.2	Sandwich (ham, cheese)	3.8
Cereal and milk	1.4	Sandwich (ham, cheese)	39.3
Cereal and milk	1.4	Sandwich (ham, cheese)	16.1
Cereal and milk (blueberries)	1.5	Sandwich (ham, egg)	11.3
Cereal and milk (vanilla soy)	2.0	Sandwich (peanut butter, jelly)	1.4
Chicken and shrimp (hibachi)	3.1	Sandwich (peanut butter, jelly)	2.4
Chicken and shrimp	2.1	Sandwich (turkey, cheese)	1.5
Chicken (teriyaki) and rice	1.2	Soup (vegetable)	29.6
Chicken wrap (mixed vegetables)	1.2	Soup (vegetable) and tortilla	63.1
Chicken wrap (vegetables)	28.8	Stir fry and quinoa	4.7
Cookie (apricot)	1.7	Toast	1.8
Eggs (boiled)	2.3	Tofu and kale	5.6
Eggs (fried) and toast	1.6	Yogurt (blackberry)	2.6
Empanadas (eggs, cheese, olives)	1.4	Yogurt and cereal	3.2
Ham and rolls	6.6	Yogurt and cereal	2.3
Lasagna	60.9	Yogurt and cereal	1.7
Lasagna	2.7	Yogurt and cereal	1.3
Oatmeal	1.7	Yogurt and cereal	1.2
Oatmeal (cranberries, bananas)	1.6	Yogurt and cereal	1.1
Pancakes (strawberry)	4.6	Yogurt (blueberries) and cereal	1.4
Pasta (tomato sauce)	1.6	Yogurt (pecan) and cereal	1.1
Peaches	6.0	Yogurt and cereal bar	6.3
Pizza	1.9		

^a^ The method quantifiable limit (MQL) was 1.1 ng/g for BPA.

**Table 2 ijerph-18-04387-t002:** Food items (*n* = 109) that contained detectable levels of TCS residues ^a^.

Food Item	TCS (ng/g)	Food Item	TCS (ng/g)
Bagel	5.2	Ham and rolls	5.3
Bagel (blueberry) and egg	1.6	Lasagna	1.2
Bagel (cream cheese)	2.2	Noodles	0.8
Bagel (cream cheese, salmon)	0.3	Noodles	0.8
Biscuit	1.4	Oatmeal	0.3
Brownie	0.5	Oatmeal	0.4
Burrito	22.1	Oatmeal	0.4
Burrito bowl (vegetable)	0.6	Oatmeal	0.9
Cake	0.3	Oatmeal	1.3
Carrots and ranch dip	1.8	Oatmeal	1.7
Cereal (dry)	0.6	Oatmeal	1.5
Cereal and milk	0.3	Oatmeal	2.1
Cereal and milk	0.3	Oatmeal (banana)	1.8
Cereal and milk	0.4	Oatmeal (banana)	6.2
Cereal and milk	0.6	Oatmeal (prunes)	3.1
Cereal and milk	1.0	Pancakes (strawberries)	2.5
Cereal and milk	1.5	Pineapples and strawberries	3.9
Cereal and milk	2.3	Pineapples and strawberries	5.4
Cereal and milk	2.9	Pizza	2.1
Cereal and milk	6.5	Pizza	3.1
Cereal and milk	0.8	Pizza (cheese)	1.8
Cereal and milk	1.4	Pizza (chicken)	1.8
Cereal and milk (banana)	0.9	Pizza (Greek)	4.4
Cereal and milk (banana)	1.3	Pizza (pepperoni)	0.9
Cereal and milk (soy)	2.8	Potato chips (barbeque)	1.7
Cereal and milk (soy, banana)	2.0	Pretzels	0.4
Cereal bar	2.8	Protein shake	1.4
Cereal bar	4.4	Protein shake	1.5
Cheese and crackers	8.7	Protein shake	2.7
Chicken (rice, vegetables)	2.9	Salad (Caesar)	0.8
Chicken (vegetables)	1.8	Salad (Caesar) and croutons	1.1
Chicken and shrimp	0.4	Salad and chicken	2.1
Chicken leg	0.8	Salad and chicken (grilled)	2.3
Chicken wrap (mixed vegetables)	0.5	Sandwich (bacon, egg)	17.4
Chicken wrap (vegetables)	1.7	Sandwich (bacon, egg, cheese)	4.0
Cookie dough	2.2	Sandwich (bologna, salami, egg)	0.6
Cookies (ginger snaps)	4.4	Sandwich (buffalo chicken)	3.4
Doughnuts (powdered sugar)	0.7	Sandwich (double cheeseburger)	1.0
Eggs (boiled)	2.0	Sandwich (egg, cheese, tomato)	4.1
Eggs (fried) and toast	2.3	Sandwich (fried ham, egg)	14.5
Eggs (scrambled) and toast	12.0	Sandwich (ham, cheese)	7.7
Eggs (scrambled, cheese, bacon)	0.6	Sandwich (ham, cheese)	10.4
Eggs and bacon	1.0	Sandwich (ham, cheese)	15.4
Empanadas	2.0	Sandwich (omelet)	2.3
Granola and granola bar	2.7	Sandwich (peanut butter, jelly)	0.6
	10.4	Spaghetti	2.1
Sandwich (peanut butter, jelly)	1.0	Stir fry (sesame noodle)	14.5
Sandwich (peanut butter, jelly)	1.5	Toast	6.6
Sandwich (tuna fish)	1.6	Yogurt	7.7
Sandwich (turkey, bologna, cheese)	9.7	Yogurt (strawberry)	2.7
Scone (pumpkin)	1.4	Yogurt (blueberries) and cereal	3.1
Soup (chicken noodle)	0.5	Yogurt (pecan) and cereal	0.9
Soup (fajita chicken)	1.0	Yogurt and cereal	0.6
Soup (vegetable)	4.4	Yogurt and cereal bar	0.8
Spaghetti	2.1	Yogurt and cereal bar	1.7
Spaghetti	1.5		

^a^ The method quantifiable limit (MQL) was 0.3 ng/g for TCS.

**Table 3 ijerph-18-04387-t003:** The adult’s estimated dietary exposures and intake doses to BPA per food item.

Food Item	Level ^a^ (ng/g)	Food Mass ^b^ (g)	Exposure (ng/day)	Intake Dose (ng/kg/day)
Sandwich (cheese, tomato)	104	132	13,770	241
Soup (vegetable) and tortilla	63.1	454	28,622	429
Lasagna	60.9	198	12,070	148
Bagel (raisin, tomato)	45.7	82.9	3789	66.2
Bagel (raisin, tomato)	42.9	97.6	4187	73.2
Sandwich (ham, cheese)	39.3	180	7086	63.3
Soup (vegetable)	29.6	868	25,678	261
Chicken wrap (vegetables)	28.8	161	4640	45.5
Salad (goat cheese, toast)	17.2	234	4016	60.2
Sandwich (ham, cheese)	16.1	116	1861	16.6
Salad and chicken	14.0	307	4302	57.3
Sandwich (ham, egg)	11.3	135	1527	13.6
Ham and rolls	6.6	156	1030	9.2
Yogurt and cereal bar	6.3	184 ^c^	1159	18.0
Peaches	6.0	242	1449	17.5
Salad (chicken barbeque)	5.6	187	1046	12.8
Tofu and kale	5.6	644	3608	36.7
Pizza (chicken)	5.1	543 ^c^	2769	48.1
Stir fry and quinoa	4.7	396	1863	18.9
Pancakes (strawberry)	4.6	255	1171	12.0
Potato chips (barbeque)	4.4	27.4	121	1.1
Burrito bowl (vegetable)	4.1	580	2378	40.9
Sandwich (ham, cheese)	3.8	176	667	6.0
Bagel (raisin)	3.6	38.2	138	2.4
Yogurt and cereal	3.2	129	414	3.2
Chicken and shrimp (hibachi)	3.1	316	978	14.7
Burrito	2.8	422	1182	9.1
Bagel (cheese, tomato)	2.7	50.9	137	2.4
Lasagna	2.7	170	458	4.8
Pizza	2.7	713	1925	24.0
Yogurt (blackberry)	2.6	218 ^c^	567	9.3
Applesauce (strawberry)	2.4	81.4	195	4.1
Sandwich (egg omelet)	2.4	75.7	182	2.4
Sandwich (egg, cheese, tomato)	2.4	305 ^c^	732	9.4
Sandwich (peanut butter, jelly)	2.4	204 ^c^	490	6.3
Yogurt and cereal	2.3	189	434	3.3
Eggs (boiled)	2.3	284 ^c^	653	10.7
Chicken and shrimp	2.1	263	553	8.3
Pizza (cheese)	2.1	248	520	5.2
Cereal and milk (vanilla soy)	2.0	254	508	6.3
Pizza	1.9	481	914	11.3
Biscuit (sausage)	1.8	110	199	1.5
Toast	1.8	18.8	33.8	0.59
Yogurt and cereal	1.7	152	258	3.7
Cookie (apricot)	1.7	17.3	29.4	0.61
Oatmeal	1.7	154	261	3.9
Bagel (blueberry, cream cheese)	1.6	84.0	134	1.5
Cake (chocolate)	1.6	83.8	134	1.5
Eggs (fried) and toast	1.6	185	295	3.3
Oatmeal (cranberries, bananas)	1.6	139	223	2.5
Pasta (tomato sauce)	1.6	312	499	7.5
Burrito (breakfast style)	1.5	122	183	1.4
Cereal and milk (blueberries)	1.5	319	479	5.4
Sandwich (egg)	1.5	150	224	3.4
Sandwich (turkey and cheese)	1.5	154	231	2.5
Cereal and milk	1.4	251 ^c^	351	5.8
Cereal and milk	1.4	167	233	3.0
Empanadas (eggs, cheese, olives)	1.4	87.4	122	1.5
Salad (Caesar)	1.4	570 ^c^	798	13.9
Sandwich (peanut butter, jelly)	1.4	408 ^c^	571	6.6
Yogurt (blueberries) and cereal	1.4	175	244	3.4
Yogurt and cereal	1.3	218	284	3.2
Cereal and milk	1.2	167 ^c^	200	3.3
Yogurt and cereal	1.2	69.6	83.5	1.3
Chicken (teriyaki) and rice	1.2	134	161	3.3
Chicken wrap (mixed vegetables)	1.2	217	260	3.2
Yogurt and cereal	1.1	186	205	1.6
Yogurt (pecan) and cereal	1.1	146	160	2.2
Cereal with milk	1.1	280	308	4.3

^a^ Only values ≥ MQL for BPA (1.1 ng/g) were used to calculate the participant’s dietary exposure and intake dose per consumed food item. ^b^ Mass of food (g) in sampling bag was recorded using a weight scale. ^c^ Mass of food was estimated using the amount of an item (in cups) recorded in the participant’s food diary.

**Table 4 ijerph-18-04387-t004:** The adult’s estimated dietary exposures and intake doses to TCS per food item.

Food Item	Level ^a^ (ng/g)	Food Mass ^b^ (g)	Exposure (ng/day)	Intake Dose (ng/kg/day)
Burrito	22.1	422	9328	72.0
Sandwich (bacon, egg)	17.4	127	2213	19.8
Sandwich (ham, cheese)	15.4	180	2777	24.8
Sandwich (fried ham, egg)	14.5	135	1959	17.5
Stir fry (sesame noodle)	14.5	238	3452	39.4
Eggs (scrambled) and toast	12.0	229	2747	31.4
Sandwich (ham, cheese)	10.4	116	1202	10.7
Sandwich (turkey, bologna, cheese)	9.7	244	2368	22.7
Cheese and crackers	8.7	177 ^c^	1540	17.7
Sandwich (ham, cheese)	7.7	176	1352	12.1
Yogurt (blackberry)	7.7	218 ^c^	1679	27.6
Toast	6.6	18.8	124	2.2
Cereal and milk	6.5	167	1084	13.9
Oatmeal (banana)	6.2	241	1496	17.1
Pineapples and strawberries	5.4	215	1163	14.0
Ham and rolls	5.3	156	827	7.4
Bagel	5.2	38.2	199	3.5
Cookies (ginger snaps)	4.4	10.6	46.6	0.67
Pizza (Greek)	4.4	488	2147	24.3
Cereal bar	4.4	163 ^c^	717	8.2
Soup (vegetable)	4.4	868	3817	38.8
Sandwich (egg, cheese, tomato)	4.1	305 ^c^	1251	16.0
Sandwich (bacon, egg, cheese)	4.0	98	392	5.2
Pineapples and strawberries	3.9	190	740	8.9
Sandwich (buffalo chicken)	3.4	650 ^c^	2210	38.4
Oatmeal (prunes)	3.1	224	693	11.5
Pizza	3.1	481	1491	18.5
Yogurt (blueberries) and cereal	3.1	175	541	7.4
Chicken (rice, vegetables)	2.9	489	1417	17.6
Cereal and milk	2.9	167 ^c^	484	8.0
Cereal and milk (soy)	2.8	254	711	8.8
Cereal bar	2.8	184 ^c^	515	8.0
Granola and granola bar	2.7	78.0	211	2.4
Protein shake	2.7	128	345	3.4
Yogurt (strawberry)	2.7	195	526	5.4
Pancakes (strawberries)	2.5	255	637	6.5
Eggs (fried) and toast	2.3	195	448	5.1
Salad and chicken (grilled)	2.3	258	593	6.8
Cereal and milk	2.3	245	564	6.4
Sandwich (omelet)	2.3	75.7	174	2.3
Bagel (cream cheese)	2.2	57.3	126	1.5
Cookie dough	2.2	117	256	3.8
Oatmeal	2.1	238	499	10.4
Pizza	2.1	713	1498	18.7
Salad (chicken)	2.1	307	645	8.6
Spaghetti	2.1	181 ^c^	380	5.0
Eggs (boiled)	2.0	284 ^c^	568	9.3
Cereal and milk (soy, banana)	2.0	530	1061	13.2
Empanadas	2.0	87.4	175	2.1
Carrots and ranch dip	1.8	185	333	4.0
Pizza (chicken)	1.8	543 ^c^	977	17.0
Chicken (vegetables)	1.8	248	447	5.1
Oatmeal (banana)	1.8	139	251	2.9
Pizza (cheese)	1.8	248	446	4.4
Potato chips (barbeque)	1.7	27.4	46.6	0.42
Chicken wrap (vegetables)	1.7	161	274	2.7
Yogurt and cereal	1.7	184 ^c^	313	4.9
Oatmeal	1.7	150	256	3.4
Bagel (blueberry, egg)	1.6	81.9	131	1.7
Sandwich (tuna fish)	1.6	173	277	3.2
Cereal and milk	1.5	338	506	5.5
Oatmeal	1.5	480	720	7.6
Sandwich (peanut butter and jelly)	1.5	126	189	2.5
Protein shake	1.5	201	302	3.0
Spaghetti	1.5	484	725	9.0
Biscuit	1.4	88.2	123	1.9
Cereal and milk	1.4	251 ^c^	351	5.8
Protein shake	1.4	331	464	4.6
Scone (pumpkin)	1.4	96.7	135	1.7
Cereal and milk (banana)	1.3	473 ^c^	615	11.3
Oatmeal	1.3	112 ^c^	145	1.9
Lasagna	1.2	170	204	2.1
Salad (Caesar) and croutons	1.1	641 ^c^	705	12.2
Soup (fajita chicken)	1.0	519	519	5.9
Cereal and milk	1.0	389 ^c^	389	4.5
Eggs and bacon	1.0	127	127	1.4
Sandwich (double cheeseburger)	1.0	291	291	2.8
Sandwich (peanut butter, jelly)	1.0	107	107	1.2
Cereal and milk (banana)	0.9	111	99.8	1.7
Oatmeal	0.9	190 ^c^	171	2.2
Yogurt (pecan) and cereal	0.9	146	131	1.8
Pizza (pepperoni)	0.9	149	134	1.4
Salad (Caesar)	0.8	285 ^c^	228	3.5
Cereal and milk	0.8	251 ^c^	201	3.7
Chicken leg	0.8	281 ^c^	225	3.7
Yogurt and cereal bar	0.8	184 ^c^	147	2.3
Noodles	0.8	491	393	4.2
Noodles	0.8	215	172	2.9
Doughnuts (powdered sugar)	0.7	166	116	1.5
Sandwich (bologna, salami, egg)	0.6	126	75.7	0.74
Cereal (dry)	0.6	58.6	35.2	0.49
Cereal and milk	0.6	111 ^c^	66.6	1.2
Sandwich (peanut butter, jelly)	0.6	204 ^c^	122	1.6
Eggs (scrambled, cheese, bacon)	0.6	297 ^c^	178	3.3
Burrito bowl (vegetable)	0.6	580	348	6.0
Yogurt and cereal	0.6	184	111	1.7
Chicken wrap (mixed vegetables)	0.5	217	108	1.3
Brownie	0.5	371 ^c^	186	2.5
Soup (chicken noodle)	0.5	351 ^c^	176	2.0
Cereal and milk	0.4	299	120	1.7
Chicken and shrimp	0.4	263	105	1.6
Oatmeal	0.4	112	44.7	0.93
Oatmeal	0.4	149	59.8	1.2
Pretzels	0.4	12.8	5.1	0.052
Bagel (cream cheese, salmon)	0.3	96.9	29.1	0.48
Cake	0.3	208 ^c^	62.4	0.63
Cereal and milk	0.3	280	83.9	1.2
Cereal and milk	0.3	150	44.9	0.74
Oatmeal	0.3	155	46.4	0.97

^a^ Only values ≥ the MQL for TCS (0.3 ng/g) were used to calculate the participant’s dietary exposure and intake dose per consumed food item. ^b^ Mass of food (g) in sampling bag was recorded using a weight scale. ^c^ Mass of food was estimated using the amount of an item (in cups) recorded in the participant’s food diary.

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
