# Peer review of "Dietary Exposures and Intake Doses to Bisphenol A and Triclosan in 188 Duplicate-Single Solid Food Items Consumed by US Adults"

_ijerph, 2021, doi:10.3390/ijerph18084387_

Round 1

Reviewer 1 Report

  • The submitted manuscript is about the determination of Bisphenol A and triclosan residue concentrations in food items consumed by adults and the estimation of dietary exposure and dietary intake doses per food item. The aim of this study is clearly defined and appropriately framed.
  • The study meets up with important concerns of the consumers and public health authorities about the presence of chemicals in foods.
  • The Introduction and the description of the state of the art are clearly articulated. 
  • Methodology is robust and the experiments are carefully designed and performed. The use of a variety of food items for the calculation of the dietary exposure to BPA and TCS residues shows experience and expertise of the authors in the subject. Real life combination of food items is a much more complicated issue that must be taken into account by scientists.          
  • The Results of the study are sufficiently analyzed and interpreted in a comprehensible way. Furthermore, authors’ conclusions are adequately supported by the results and are clearly related to the literature, which is important in order to help the reader determine what is to be learned and what can be speculative.
  • The manuscript is very well written concerning the English grammar, style and syntax.
  • I have no major comments regarding this manuscript which is of publishable quality.
  • A minor comment/suggestion for the authors’ consideration. Public health concerns about the use of such chemicals are not pinpointed sufficiently in the study (lines 27, 54, 61-67), at least in my opinion. Triclosan is banned for use in the manufacture of plastics intended to come into contact with food in many countries, for example in the European Union (SCCS Opinion on Triclosan, ISBN 978-92-79-12484-6). Legal aspects concerning such chemicals should be highlighted in a more commentary way to help the inexperienced reader of the manuscript understand restrictions regarding foods.

Author Response

Reviewer 1 comments - A minor comment/suggestion for the authors’ consideration. Public health concerns about the use of such chemicals are not pinpointed sufficiently in the study (lines 27, 54, 61-67), at least in my opinion. Triclosan is banned for use in the manufacture of plastics intended to come into contact with food in many countries, for example in the European Union (SCCS Opinion on Triclosan, ISBN 978-92-79-12484-6). Legal aspects concerning such chemicals should be highlighted in a more commentary way to help the inexperienced reader of the manuscript understand restrictions regarding foods.

Response: We have revised the following sentences in the Introduction section to address the missing public health concerns information (lines 28-31) Bisphenol A (BPA) is a synthetic phenol with over 2 billion pounds used each year in the United States (US) [1,2]. Research has indicated that BPA is an endocrine disruptor and may be causing adverse health effects (e.g., reproductive, neurological, and obesogenic) in exposed humans [1,2]. (Lines 56-59) - Triclosan (TCS) is a chlorinated phenol with more than 1 million pounds produced annually in the US [20,21,22]. Studies have shown that TCS is an endocrine disruptor and may be impacting the health (i.e., thyroid and reproductive) of exposed humans [20,21]. We have also revised the following sentence (line 59) -  This phenol is not currently used in any food packaging materials (e.g., boxes, cans, bags, or cartons) in the US [22] and in countries from the European Union [21].

Reviewer 2 Report

It is my pleasure to be able to review the manuscript entitled "Dietary Exposures and Intake Doses to Bisphenol A and Triclosan in 188 Duplicate-Single Solid Food Items Consumed by US Adults" by Morgan and Clifton. The paper is well written, results interesting and conclusions clear. However, this article is very similar to one cited by the authors, "Exposure to Triclosan and Bisphenol Analogues B, F, P, S and Z in Repeated Duplicate-Diet Solid Food Samples of Adults", as data are driven from the same samples and the same compounds are analyzed. So these are my concerns:

-I am not sure what is the difference between this article and the article "Exposure to Triclosan and Bisphenol Analogues B, F, P, S and Z in Repeated Duplicate-Diet Solid Food Samples of Adults", as datasets are the same and compounds studied are the same. Did you perform additional analysis? Please explain. The only explanation I found is "For this current work, we have quantified BPA and TCS residue levels occurring in the 188 duplicate-diet solid food samples that contained only a single food item." in page 3, lines 103-104

-BPA and TCS are known endocrine disruptors, there is an enormous amount of literature that support this idea. This is why it is intriguing why the authors say that these two chemicals are "suspected endocrine disruptors" (page 1 line 27, page 2 line 54)

Author Response

Reviewer 2 (first comment) - I am not sure what is the difference between this article and the article "Exposure to Triclosan and Bisphenol Analogues B, F, P, S and Z in Repeated Duplicate-Diet Solid Food Samples of Adults", as datasets are the same and compounds studied are the same. Did you perform additional analysis? Please explain. The only explanation I found is "For this current work, we have quantified BPA and TCS residue levels occurring in the 188 duplicate-diet solid food samples that contained only a single food item." in page 3, lines 103-104

Response: We have revised the following paragraph providing clarification on the work performed in our two previous articles and in this current manuscript - “In our previous work from the Ex-R study [9,31], we quantified the concentrations of BPA and TCS in 776 duplicate-diet solid food samples of 50 adults and estimated their maximum dietary exposures and dietary intake doses to these two chemicals. However, at that time it was unclear what were the specific consumed solid food items (e.g., cheeseburger, salad, and pizza) that were likely substantially contributing to the dietary exposures of the Ex-R adults to BPA or TCS. In a further analysis of the Ex-R study data, the participants’ food diary records were used to determine that 188 of the collected 776 duplicate-diet solid food samples contained only a single food item. For this current work, the main objectives were to 1) determine BPA and TCS residue concentrations in the 188 duplicate-single solid food items consumed by adults and 2) estimate dietary exposure and dietary intake doses per food item. This research is important as we are unaware of any published study that has reported BPA or TCS residue concentrations in duplicate-single solid food items consumed by adults in their everyday settings.

Reviewer 2 (second comment) - BPA and TCS are known endocrine disruptors, there is an enormous amount of literature that support this idea. This is why it is intriguing why the authors say that these two chemicals are "suspected endocrine disruptors" (page 1 line 27, page 2 line 54)

Response: We agree. We have revised the following sentences based on the reviewer’s comment (& reviewer 1 comments) - Bisphenol A (BPA) is a synthetic phenol  with over 2 billion pounds used each year in the United States (US) [1,2]. Research has indicated that BPA is an endocrine disruptor and may be causing adverse health effects (e.g., reproductive, neurological, and obesogenic) in exposed humans [1,2]. (Lines 56-59) - Triclosan (TCS) is a chlorinated phenol with more than 1 million pounds produced annually in the US [20,21,22]. Studies have shown that TCS is an endocrine disruptor and may be impacting the health (i.e., thyroid and reproductive) of exposed humans [20,21].

Reviewer 3 Report

The proposed manuscript is a well documented study. The methodology seems appropriate and the study framework have been developed according to the state of art. I id not found any significant drawbacks. I consider this paper as suitable for publication in the present form. 

Author Response

Reviewer 3 comments - The proposed manuscript is a well documented study. The methodology seems appropriate and the study framework have been developed according to the state of art. I id not found any significant drawbacks. I consider this paper as suitable for publication in the present form. 

Response: The reviewer did not request any changes to this manuscript.

Round 2

Reviewer 2 Report

It is my pleasure to be able to review the manuscript entitled "Dietary Exposures and Intake Doses to Bisphenol A and Triclosan in 188 Duplicate-Single Solid Food Items Consumed by US Adults" by Morgan and Clifton. The paper is well written, results interesting and conclusions clear. With the clarifications added, I am ok with accepting the paper now.